# The Effect of the Corporate Social Responsibility of Franchise Coffee Shops on Corporate Image and Behavioral Intention

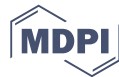

**Jae-Bin Cha [1] and Mi-Na Jo [2,*]**

[1]   Department of Public Health Administration, Kyungmin University, Euijeongbu 11618, Korea; jbcha004@kyungmin.ac.kr

[2]   Division of Hotel & Tourism, The University of Suwon, Suwon 18323, Korea

*   Correspondence: jomina@suwon.ac.kr; Tel.: +82-31-229-8308

**Abstract:** This study investigated the effects of the corporate social responsibility (CSR) of franchise coffee shops on their corporate image and their customers' behavioral intention. From 5 October 2017 to 26 October 2017, a total of 300 survey questionnaires were distributed, out of which 285 were analyzed. Accordingly, among the dimensions of CSR, only the economic, discretionary, and legal responsibilities had significant effects on corporate image. Moreover, corporate image had a significant effect on customers' behavioral intention. Therefore, the CSR of franchise coffee shops is an important antecedent of corporate image, while corporate image is an important antecedent of behavioral intention. With respect to the pyramid of CSR, economic responsibilities should be first fulfilled prior to other social responsibilities. In addition, CSR programs will become the core task of corporate sustainable management because CSR can potentially influence consumers to have a positive behavioral intention and can result in product preferences by conveying a particular corporate image.

**Keywords:** CSR; corporate image; behavioral intention

---

## 1. Introduction

Despite the global economic recession, the coffee industry in Korea has witnessed continued growth in the last decade. Korea Customs Service reported that one adult consumes 512 cups of coffee a year in 2017, thus indicating coffee is consumed more often than kimchi or rice. In 2006, a total of 92,000 tons of coffee volume was imported, and this significantly increased to 159,000 tons in 2016, the highest amount in the country's history [1]. Franchise coffee brands are also making rapid growth along with the popularity of coffee. In particular, after Starbucks Korea opened its first chain in 1999 through a joint venture between Shinsegae and Starbucks International, global coffee brands such as Coffee Bean and Caffe Pascucci, as well as the local brands including Hollys Coffee, Ediya Coffee, Tom N Toms Coffee, Caffé Bene, Angel-in-us Coffee, and A Twosome Place grew quickly in the market [2]. The industry of franchise coffee has recorded over 25% growth every year, with the number of chains reaching 50,000 shops as of January 2018—an eight-fold increase from 6000 shops in 2008 [3]. In addition, the top-selling brands, such as Starbucks, Caffe Bene, Coffee Bean, Hollys Coffee, and Angel-in-us Coffee, have over 3000 chains [3].

However, such a large number of stores has led to fierce competition among and dwindling profitability of these franchise coffee brands. As a result, the coffee industry is confronted with intense competition and extreme changes, and hence the emerging importance and need for corporate philanthropy so franchise coffee brands can have a differentiated marketing vehicle aside from their

usual profit-chasing activities. Consequently, a number of franchise coffee brands have sought to establish a positive brand image by implementing corporate social responsibility (CSR) in order to enhance employees' pride and consumer preference for their brand.

As previously mentioned, a number of brands are implementing diverse programs. Consumers prefer products from companies that are committed to CSR activities when products are engaged in tough competition [4].

CSR refers to the obligations of entrepreneurs to pursue politics, to make decision, or to follow lines of actions which are desirable in terms of the objectives and values of society [5].

The concept of CSR was first introduced in the 1930s; it then became more substantial in the 1960s given the changes in social values during that time, which resulted from expanding companies and their social influence and conflicts. In 1990s, the concept became referred to as corporate sustainability and responsibility to describe the pursuit of environmentally, financially, and socially sustainable corporate growth through healthy and responsible corporate management.

In the coffee industry, CSR programs started because of the collapse of the International Coffee Agreement in 1989, during which prices plunged dramatically and reached their lowest point in 2002 [6]. Attention was then focused on the role of large multinational corporations because of their important position in what has become a buyer-driven commodity chain. By the mid-1990s, large multinational coffee corporations started to experience major pressure from non-governmental organizations (NGOs). Such trends led to the overall move towards CSR. Starbucks and P&G buy relatively small amounts of fair trade coffee. Kraft and SLDE, as larger mainstream players, have started buying certified coffee and paying extra money for higher quality and certification activities; whereas Nestle, which has production plants in developing countries, directly procures coffee from farmers at a considerably higher price than they would have received otherwise [7]. The website of Starbucks in 2016 also contains a responsibility section. Starbucks explains that they offer high-quality, ethically purchased, and responsibly produced products. They also invest in opportunities through education, training, and employment to create more opportunities and minimize their environmental effects. Finally, they position Starbucks as a place of public conversation and elevate civic engagement by rendering service and promoting voter registration so as to encourage service and citizenship [8].

In Korea, CSR is more proactive as it has become a critical part of corporate innovation and corporate competitiveness that enhances a company's sustainable growth. With respect to corporate investment in CSR, an amount totaling to 0.6 billion dollars was recorded in 2000, but increased to 2.5 billion dollars in 2015, thus marking a growth in both quality and quantity. In addition, the "Study on Company CEOs' Perception on CSR" suggests that the scale of CSR will remain as is (50.7%) or will likely increase to 35.1%, despite the ongoing economic recession [9].

While the CSR has become a critical agenda in the western countries in the mid-1990s, its importance increased in East Asian countries in the last 10 years [10]. Lee [10] compared CSR in four key countries in East Asia that focus on capitalist growth, namely, Japan, Korea, China, and Taiwan. Accordingly, the normative, the conspicuous, the enlightened, and the bystander CSR have been established in Japan, Korea, China, and Taiwan, respectively. Normative CSR conforms to the market principles and has been led by firms in Japan. Conspicuous CSR is based on the owners of business groups in Korea. Enlightened CSR is driven by the government-initiated attempts to encourage public companies based on a strong descriptive structure in China. Finally, bystander CSR has surfaced, one form that has been initiated by the state but ignored by the corporate sector in Taiwan.

While the corporate competition has become ever fiercer, the efforts to obtain higher respect has also turned into a part of the strategic approach for companies to establish positive image, and this is one of the main reasons why CSR is influential in corporate image building [11,12]. As a positive approach to improve corporate awareness and maintain a positive corporate image in the market, CSR enhances a company's competitiveness. Corporate image is enhanced by more positive consumer preference and behavior, hence reinforcing behavioral intention [13,14]. When customers' behavioral intention is positive, then corporate performance is likewise further advanced.

Indeed, CSR positively influences corporate image, and such company's products and brands have an impact on customers' behavior throughout their behavioral intention [11,15–17]. Given that the difference in technology and products among companies has become less significant, a company's CSR activities have emerged as a critical device to persuade consumers. In addition, customers' responses change in terms of company's CSR approach and communication method; therefore, a CSR approach is vital in a company [13,18,19]. Drumwright [20] also stated that a company decides on the CSR approach they would take and who would sponsor it, and that CSR is more effective when there is a high relevance with the brand.

Thus, companies should strategically decide which CSR activity they would take on the marketing end. In this case, the relevance between the CSR activity and the brand should be considered as it is more effective in enhancing consumer's purchase intention [21]. In addition, CSR activity is not only related to cost increase; in fact, it is a type of investment similar to marketing that enhances corporate image and purchase intention among consumers [22–24].

Recently, CSR has become an essential part to become a respected and/or sustainable company. The importance of CSR is considered given the rapidly changing business environment. For instance, the U.S. has implemented the Sarbanes-Oxley Act of 2002 and the UN has adopted the Global Compact in 1999, both reinforcing the importance of ethical management and CSR. In particular, the International Organization for Standardization (ISO) legislated ISO 26000 in terms of CSR [25].

In Western countries, ample research has been conducted on CSR in the coffee industry. However, in Korea, one of the East Asian countries, only a limited amount of research has been focused on CSR in coffee companies [26–28].

Therefore, this research intended to study the impact of CSR of the top four franchise coffee companies in Korea on their corporate image and customers' behavioral intention. Considering the increase in coffee awareness, the expansion of the coffee market, and the rapid growth in coffee culture, this study will be relevant as it focuses on the local and international franchise coffee brands that conduct business in Korea.

## 2. Theory and Hypotheses

### 2.1. Corporate Social Responsibility

While the company's foremost goal is to pursue profitability and financial performance through effective business activities, the modern companies also have the responsibility to make a healthier and proper society. The companies agree upon the importance of CSR, and due to the legislation of ISO 26000 in 2010, maintaining a company's corporate social responsibility has become essential, rather than an option, and is now a shared interest among companies worldwide [29].

ISO 26000 was developed by ISO (the International Organization for Standardization) and it is a global standard for CSR (Corporate Social Responsibility). ISO 26000 provides guidance rather than requirements on how businesses and organizations can operate in a socially responsible way to help clarify what social responsibility is, help businesses and organizations translate principles into effective actions, and share best practices relating to social responsibility.

The term CSR was first formalized by Bowen [5], who argued that "it refers to the obligations of businessmen to pursue those politics, to make those decisions, or to follow those lines of actions which are desirable in terms of the objectives and values of society". A decade later, several authors, including from references [17,30–32], undertook further development of the concept [33]. McGuire [32] states "CSR is not only related to the economic and legal responsibility of company on society, but also its responsibility over the society," and Wartick and Cochran [34] defines CSR as an integration of social responsibility, social responses, and a social agenda. In addition, Eells and Walton [35] insist that a company ought to perceive social responsibility from the viewpoint of a problem that occurs as a result of the business and from the viewpoint of the ethical principle between its relationship with the society. Afterwards, a company should come up with proper solution, while conveying its social

responsibility by adhering to its ethics. Moreover, Sethi [36] defines CSR as a business practice that can actualize economic and legal responsibilities, social norms and values, and social expectations and harmony.

Carroll [37] categorizes social responsibilities into four sectors: economic responsibilities, legal responsibilities, ethical responsibilities, and discretionary responsibilities. Economic responsibilities are the first and foremost social responsibility of a business. Because the business institution is the basic economic unit in our society, it has a responsibility to produce goods and services that society wants and to sell them at a profit. All other business roles are predicated on this fundamental assumption. Legal responsibilities mean that society expects business to fulfill its economic mission within the framework of legal requirements. Ethical responsibilities are not necessarily codified into law but nevertheless are expected of business by society's members. Discretionary responsibilities are voluntary activities and the decision to assume them is guided only by a business's desire to engage in social roles. Among the four categories of social responsibility defined by Carroll [37], economic, legal and ethical responsibilities are the areas that must be carried out by the company, while discretionary responsibility is a voluntary option.

Carroll [38] first presented his CSR model as a pyramid. In this pyramid, economic responsibility is the basic foundation. Without the fulfillment of the basic foundation, the upper category cannot be carried out. In other words, philanthropic responsibility practices without the successful fulfillment of economic, legal, and ethical responsibilities will be doubted in terms of their integrity, thus failing to incur a positive result. He suggested that four kinds of social responsibilities constitute total CSR. Petkus and Woodruff [39] defined CSR as a company's commitment to minimize or eliminate any harmful effects and maximize its long-term beneficial impact on society—an interpretation likewise adapted by Mohr, Webb, and Harris [24]. Mohr, Webb, and Harris [24] defines CSR as a business practice conveyed to fulfill social demands. In fact, CSR is a commitment; it is divided into passive practices, such as an economic and legal responsibility to minimize or eliminate any harmful effects on society, and into voluntarily and proactive social practices including ethical and charitable responsibilities that do not have any binding force.

In short, CSR refers to socially beneficial activities, contribution, sponsoring activities, and environmental protection activities on a narrower scope, as well as an economic responsibility for profit maximization, a legal responsibility, an ethical responsibility for ethical management, and voluntary responsibility for social contributions on a larger scale [18,40]. Since CSR has the power to positively influence a company's sustainability, competition, and performance, it is useful for the companies to utilize it as part of a more active and long-term perspective. For that reason, this study refers to CSR as a company's fulfillment of its economic, legal, ethical, and charitable responsibilities [37].

Brown and Dacin [12] studied the impact of CSR and showed that a high CSR grade led to a higher evaluation of the company, while corporate evaluation was positively related to product evaluation. Murray and Vogel [41] said that people had more positive attitudes toward the company and more positive behavioral intentions when the company's prosocial programs were described. Handelman and Arnold [42] found CSR to have a strong impact on support. Even though a store was described as being strong on store image attributes like its assortment of goods, prices, and convenience of location, this was useless unless the store was also supportive of families, local charities and product made locally. This means that being socially irresponsible overshadowed the traditional criteria that consumers use to select retailers. Those results indicate that information on CSR can have a significant impact on behavioral intentions as well as their evaluations of products and companies [24].

## 2.2. Corporate Image

Barich and Kotler [43] use the term "image" to represent the sum of beliefs, attitudes, and impressions that a person or group has of an object. The object may be a company, product, brand, place, or person. Gardner and Levy [44] introduced the concept of "image" and discussed corporate image, product image, and brand image, while Barich and Kotler [43] added the fourth type of image,

which is a company's marketing image. Hence, different definitions are given to corporate image, including that it is a person's perception [45,46], a mental picture, or a portrait of a firm [47,48] incorporates evaluations, feelings, and attitudes toward a company into their conceptualizations of a company's image [43,49,50]. Even though specific definitions of corporate image differ, authors agree that it exists in people's minds and that there is not a shared corporate image for any given company. All the frameworks that were proposed regarding corporate image [43,51,52] insist that a company has multiple audiences or constituencies, namely, consumers, the business community, government, news media, and employees [12]. Corporate image describes how the public thinks of the company's goodwill toward society, employees, customers, and other stakeholders [43].

Corporate image has a cumulative effect on customer satisfaction and dissatisfaction [53–56]. When services are difficult to evaluate, corporate image is believed to be an important factor influencing perceptions of quality, customers' evaluation of satisfaction with the service, and customer loyalty [57]. Especially when product technologies and technical standards have been enhanced and standardized, there is a limit to product difference. For that reason, corporate image has become a critical factor. The companies are putting major efforts into enhancing their corporate images through diverse advertisements and corporate social responsibility activities. Once a positive corporate image has been sustained for a long period of time, that company would have a positive reputation as a result, and such an image will likely lead to huge profits. In addition, a positive corporate image will influence brand knowledge and relief, triggering favorable consumer brand behaviors and intention. The overall corporate evaluation exhibits an influence on product evaluation [12]. For that reason, the companies should focus on their corporate image improvement strategy for business efficiency.

While there are many opinions as to the factors that impact corporate image, the consensus is that CSR is one of the critical elements [11,12,58]. CSR is one of the particular features of a company that are relevant to major social issues. For that reason, it is a concept that is deeply engaged with environmentally friendly approaches, respect towards diversity in terms of employment, community spirit, and the company's cultural events or sponsorship. It is indeed a positive strategic element in forming a good corporate image.

Company leaders perceive CSR as a critical measure in the success or failure of cultivating a positive corporate image [59]. When customers purchase a product, they consider positive images first, which means that this factor could induce the sales trigger. On the other hand, the companies with a negative image have a hard time converting this into a positive image despite abundant investment efforts, and their products will be negatively assessed [60] While corporate image is not the decisive factor in purchase decisions, it serves as a role in pre-sales decisions by enhancing consumer awareness and understanding level regarding the company or their products [61]. For that reason, companies could promote their corporate image through diverse social responsible programs, and ultimately purse profit maximization [62].

*2.3. Behavioral Intention*

Behavioral intentions are indications of whether a visitor to a program or facility will return at a future date. Reference [63] stated in their "Theory of Reasoned Action" that behavioral intention refers to "customers' expression of will for their future action." The theory of reasoned action postulates that behavior can be predicted from intentions that correspond directly to that behavior [64]. Fishbein and Manfredo [65] conclude that correspondent intentions are very accurate predictors of most social behaviors. And perceptions of high quality positively affect intended behavior [66,67].

Behavioral intention is a concept that includes satisfaction as well as intentions to return or provide recommendations to others, which refers to their will to re-purchase or re-use certain products [66]. In addition, it could be used as a prediction factor for certain actions [63]. The will for an action is a critical element in forecasting customer actions, and it can refer to a behavior formed after using a particular product or service, and to the customer's will for future action [68]. The will for an action is a emotional response of an individual, which is very critical in the customer's decision making

process when purchasing a particular product or target, and it explains their amount of satisfaction as a whole [69].

There is a relationship between a company's CSR activities and consumer patronage of its products, while awareness of the CSR is associated with an increased desire to buy a company's products [70]. Turban and Greening [71] described CSR as a significant driver of a company's attractiveness to potential employees. University undergraduates are sensitive to CSR issues and they increase their likelihood of investing in a company based on its CSR activities [70].

When expectations are exceeded by actions, firms are rewarded with more positive attitudes and an increase in positive behavioral intentions [22,67]. But only proactive social initiatives can exceed baseline expectations and lead to more positive beliefs, attitudes, and intentions [22]. In the case of reactive initiatives, context provides cues of firms' motivations, prompting consumers to question company actions and choosing to lower their CSR evaluations of the company [72]. In contrast, under conditions of proactive initiatives, the context and motivations of companies are more ambiguous. As such, a company should manage their CSR activities properly to avoid increasing skepticism and fostering negative attitudes and beliefs towards firms [73].

### 2.4. The Research Model

The research model based on the hypotheses is presented in Figure 1:

**H1.** *Economic Responsibilities are positively related to Corporate Image.*

**H2.** *Ethical Responsibilities are positively related to Corporate Image.*

**H3.** *Legal Responsibilities are positively related to Corporate Image.*

**H4.** *Discretionary Responsibilities are positively related to Corporate Image.*

**H5.** *Corporate Image is positively related to behavioral intention.*

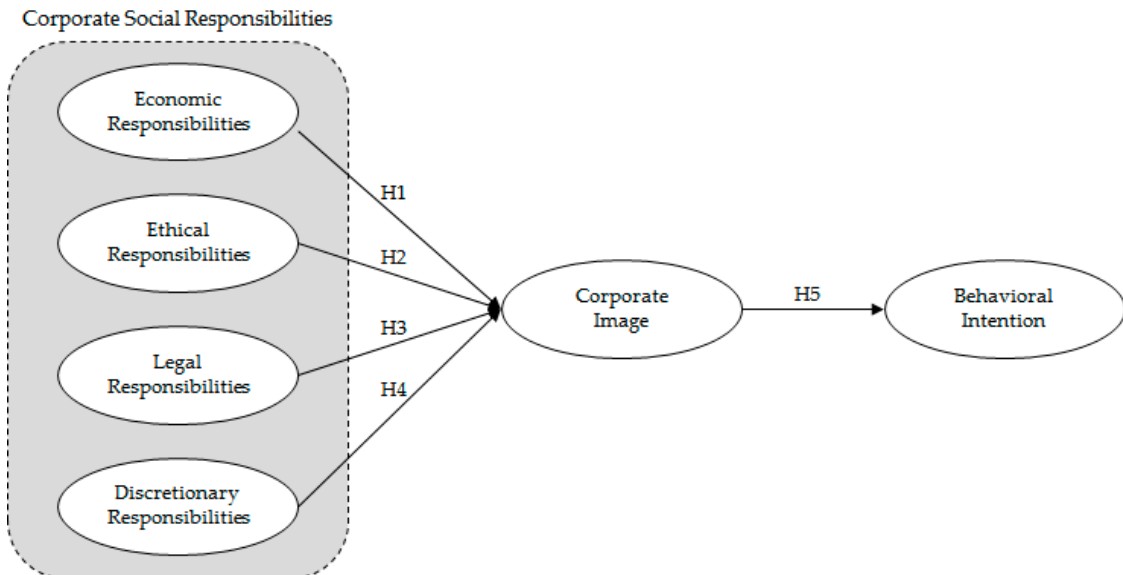

**Figure 1.** The research model.

Social responsibility refers to a company's responsibility for improving broader society, in addition to the corporate sector's motivation for profitability [5]. The corporate image is a total sum of an individual or a group's belief, behavior, and impressions [43], which means that it is a consumer's overall impression of a company.

In a service industry like the franchise coffee brands where a direct relationship with the customer is critical, the customer's awareness of the company's public behavior influences the credibility of the company and the customer's overall quality assessment [74].

Moreover, CSR is now positioned as a more influential factor in establishing a positive corporate image. From the perspective of corporate substantiality management, CSR is beneficial in acquiring validity from diverse stakeholders including shareholders, employees, customers, and regional communities, as well as benefitting the long-term growth of the company. Accordingly, CSR has a positive effect on employees' job satisfaction and their concentration on the organization's tasks [24]. CSR activity enhances not only a company's competitiveness but also its brand image, thus positively influencing a customer's behavior and purchase intention [12].

A company's CSR activities establish a positive image surrounding the company [75], and the image of a shop and their socially responsible activities have positive relevance [42]. In addition, diverse CSR activities promote a positive corporate image, ultimately contributing to a profit increase for the company [76–78].

In other words, diverse CSR activities of a company are expected to establish a positive image for the company. Based on these findings, this research study has established the following hypotheses. This study has formed the constituent factors of a CSR activity based on the four-phase model of Carroll [38], they are: economic, legal, ethical, discretionary and responsibility.

**H1.** *Economic Responsibilities are positively related to Corporate Image.*

**H2.** *Ethical Responsibilities are positively related to Corporate Image.*

**H3.** *Legal Responsibilities are positively related to Corporate Image.*

**H4.** *Discretionary Responsibilities are positively related to Corporate Image.*

A corporate image is a convergence of customer's overall belief, behavior, and knowledge of a company [43]; in other words, it could be defined as a comprehensive impression of a consumer regarding a company. Barich and Kotler [43] use the term "image" to represent the sum of beliefs, attitudes, and impressions that a person or group has of an object. The object may be a company, product, brand, place, or person.

If a company conducts a CSR activity, a consumer will display positive behavior regarding the company's brand or the product [79,80], and they will positively evaluate the product, which will have a meaningful impact on their purchase intention [16,19,22,81]. In addition, Assael [15] stated that customers evaluate a company based on their information and experience of their product, and establish a belief of their own of a company based on their information and experience. They also stated that such a belief will be either positive or negative towards the brand, which will impact their purchase intention or purchase behavior. Also, Lafferty and Goldsmith [82], Madrigal [83], Mohr, and others [24] have proved in their studies that companies that carry out excellent social responsibility activities have higher consumer purchase intentions than those that do not, implying that social responsibility triggers a chain of positivity, which results in positive evaluation of products [74].

A positive corporate image is known to have an impact on the maintenance of the relationship between a customer and a company. Kang JW and Namgung Y [84] stated that when franchise coffee companies are recognized as having a positive and friendly image, their customers have positive behaviors, and therefore, this should be leveraged in a company's strategy development. In the same context, Han SJ [85] said that a favorable evaluation of a franchise coffee brand's corporate image will form positive brand consumer behavior, which acts as a purchase trigger for the customers, playing a pivotal role in their desire to revisit the store and their will to recommend it to others.

Likewise, corporate image is a pivotal factor that determines customers' behavior and attitudes towards a company; therefore, positive corporate image is assumed to have a positive impact on customers' perceptions of the brand and their behavioral intentions. Based on this, the hypothesis below has been set.

**H5.** *Corporate Image is positively related to behavioral intention.*

## 3. Methodology

### 3.1. Data Collection and Participant Characteristics

For this investigation, an onsite survey has been conducted based on a self-administered survey targeting customers at franchise coffee shops in Seoul and Gyeonggi area from 5 October, 2017 to 26 October, 2017. A total of 300 copies were distributed and every one of them has been retrieved. Except for 15 copies, which had either inconsistent or insincere answers, all of the copies were used for the final analysis. Frequency Analysis, Confirmatory Factor Analysis (CFA), Reliability Analysis, Correlation Analysis, and the Structural Equation Model (SEM) were adopted for the study. As for the sample, the gender divide was 47.0% of male respondents (n = 134) and 53.0% (n = 151) of female respondents. In terms of age groups, 26.3% (n = 75) of respondents were 20–29 years of age, 31.2% (n = 89) were 30–39 years of age, 33.7% (n = 96) were 40–49 years of age and 8.8% (n = 25) were over 50 years of age. In terms of the company with the highest CSR corporate image, 47.0% (n = 134) chose Starbucks, 20.4% (n = 58) chose Caffe Bene, 18.9% (n = 54) chose Coffee Bean, and 13.7% (n = 39) chose Tom n Toms Coffee.

### 3.2. Measurement Scales

Based on the practical background and preceding research, each constituent concept has been adopted and revised for the survey questionnaires to fit this research. Corporate social responsibility has been categorized using four factors in accordance with the investigation studies by Maignan and Ferell [19] and Carroll [37]: economic, legal, ethical, and discretionary responsibilities. The social responsibility that the customers demand from a company may differ in accordance with the economic development of a company, awareness of democratic principles, and culture. Likewise, this study has adopted 16 items based upon the scale for measuring the corporate social responsibility activities of Korean corporations developed by Park JC [86]. The corporate image in this case refers to a particular image, behavior, and belief that the customer, public, or relevant institutions have about the company rather than towards the product, brand name, or the company itself. The corporate image was measured in terms of seven items, implying the measurement of Winters [58], and Brich and Kotler [43]. A customer's behavioral will is defined as a will to re-purchase or re-visit as an overall concept that covers their satisfaction, willingness to revisit the store, and willingness to recommend the brand to others. Each measured item was based on a five point Likert scale from "do not agree at all" to "agree very much".

## 4. Results

### 4.1. Reliability, Validity, and Common Method Bias

The reliability and validity of the resulting measurement scales were assessed. First, the reliability of the constructs was assessed using Cronbach's alpha coefficient (see Table 1). The reliability coefficients for the constructs ranged from 0.695 to 0.914, which is considered satisfactory [87]. Confirmatory factor analysis (CFA), using AMOS 22.0 software, was used to evaluate the convergent and discriminant validity of the measurement items.

After the confirmatory factor analysis, a two-step analysis approach that analyzes the measurement model was conducted [88,89].

**Table 1.** Scale items and construct evaluation.

| Construct | | λ | α | CR | AVE |
|---|---|---|---|---|---|
| Economic Responsibilities | The product quality (or service) seems to be improving constantly. | 0.524 | 0.695 | 0.775 | 0.465 |
| | Seems to have a system that reacts to customer complaints. | 0.644 | | | |
| | Contributes to national economic growth through profit maximization. | 0.569 | | | |
| | Puts much effort into employment. | 0.686 | | | |
| Ethical Responsibilities | Has high overall ethics standards and protocols. | 0.683 | 0.814 | 0.855 | 0.596 |
| | There are no excessive ads or false ads. | 0.641 | | | |
| | Conducts transparent business. | 0.699 | | | |
| | Carries out fair transactions with business partners. | 0.691 | | | |
| Legal Responsibilities | Products meet legal standards. | 0.693 | 0.834 | 0.870 | 0.626 |
| | Contributes to social welfare systems as mandated by law. | 0.770 | | | |
| | Seems to fulfill the responsibilities indicated on their contracts with other partners. | 0.771 | | | |
| | Management seems to put effort into ethical business management, complying with the product related regulations. | 0.754 | | | |
| Discretionary Responsibilities | Encourages collaboration of business with the regional community and other institutions. | 0.765 | 0.819 | 0.863 | 0.612 |
| | Sponsors sports and cultural events. | 0.693 | | | |
| | Encourages charity services supporting regional communities. | 0.705 | | | |
| | Gives back to society. | 0.712 | | | |
| Corporate Image | Have good impressions of the company. | 0.766 | 0.914 | 0.924 | 0.671 |
| | Perceive the company in a positive way. | 0.757 | | | |
| | I like the company. | 0.854 | | | |
| | The company is trustworthy. | 0.802 | | | |
| | The company puts much effort into customer satisfaction. | 0.736 | | | |
| | The company is an excellent company. | 0.802 | | | |
| | The company is beneficial to society. | 0.742 | | | |
| Behavioral Intention | I will try to revisit the shop. | 0.785 | 0.907 | 0.919 | 0.654 |
| | This company will be on my priority list. | 0.828 | | | |
| | I will still visit the brand even if their prices increase. | 0.710 | | | |
| | I will come to this brand even if there is another brand nearby. | 0.833 | | | |
| | I will provide positive comments about this brand to others. | 0.857 | | | |
| | I will actively recommend this brand to my family or acquaintances. | 0.830 | | | |

χ2(302) = 252.131; *p* > 0.05, GFI = 0.944, AGFI = 0.920, NFI = 0.955, RFI = 0.935, TLI = 1.000, CFI = 1.000, RMSEA = 0.000, RMR = 0.023

The measurement model fits well with the data, as seen by the fit statistics for the model (χ2(302) = 252.131; *p* > 0.05, GFI = 0.944, AGFI = 0.920, NFI = 0.955, RFI = 0.935, TLI = 1.000, CFI = 1.000, RMSEA = 0.000, RMR = 0.023). Across our measurement models, the factor and item loadings all exceeded 0.5, with all t-values greater than 2.58, demonstrating the convergent validity among our measures. All the measures provided high reliability, with composite reliabilities ranging from 0.775 to 0.924 (see Table 1). We investigated the qualification for discriminant validity among variables,

as suggested by Fornell and Larcker [90]. The Average Variance Extracted (AVE) values were all larger than the squared correlation between the construct and any others (see Table 2). Overall, our constructs showed good measurement properties.

**Table 2.** Scale items and construct evaluation.

|  | 1 | 2 | 3 | 4 | 5 | 6 |
|---|---|---|---|---|---|---|
| 1. Economic Responsibilities | 0.465 |  |  |  |  |  |
| 2. Ethical Responsibilities | 0.560 ** | 0.596 |  |  |  |  |
| 3. Legal Responsibilities | 0.571 ** | 0.638 ** | 0.626 |  |  |  |
| 4. Discretionary Responsibilities | 0.524 ** | 0.533 ** | 0.617 ** | 0.612 |  |  |
| 5. Corporate Image | 0.613 ** | 0.634 ** | 0.657 ** | 0.571 ** | 0.671 |  |
| 6. Behavioral Intention | 0.521 ** | 0.552 ** | 0.593 ** | 0.527 ** | 0.806 ** | 0.654 |
| Mean | 3.074 | 2.961 | 3.115 | 2.782 | 3.134 | 3.008 |
| SD | 0.594 | 0.647 | 0.712 | 0.665 | 0.726 | 0.797 |

Note: The number in the diagonal is the AVE; ** $p < 0.01$.

### 4.2. Hypothesis Test

A SEM method was used to examine the hypothesized connections. The fit indices of the research model are generally acceptable (Table 3). Thus, the research model ($\chi 2(308) = 234.668; p > 0.05$, GFI = 0.947, AGFI = 0.925, NFI = 0.955, RFI = 0.941, TLI = 1.000, CFI = 1.000, RMSEA = 0.000, RMR = 0.022) is deem to fit well with the collected data.

**Table 3.** Scale items and construct evaluation.

| Hypotheses | Standardized Coefficient | SE | CR | Results |
|---|---|---|---|---|
| H1 Economic Responsibilities → Corporate Image | 0.317 ** | 0.132 | 3.134 | Supported |
| H2 Ethical Responsibilities → Corporate Image | 0.176 | 0.115 | 1.737 | Not Supported |
| H3 Legal Responsibilities → Corporate Image | 0.216 * | 0.119 | 1.974 | Supported |
| H4 Discretionary Responsibilities → Corporate Image | 0.218 * | 0.106 | 2.362 | Supported |
| H5 Corporate Image → Behavioral Intention | 0.867 *** | 0.075 | 12.936 | Supported |

$\chi 2(308) = 234.668$; $p > 0.05$, GFI = 0.947, AGFI = 0.925, NFI = 0.955, RFI = 0.941, TLI = 1.000, CFI = 1.000, RMSEA = 0.000, RMR = 0.022

Note: * $p < 0.05$, ** $p < 0.01$, *** $p < 0.001$.

Figure 2 and Table 3 present a summary of the tested hypotheses. All structural coefficients of connections among the factors were significant.

We tested the hypothesis 1, the economic responsibility fulfillment left meaningful impact on the corporate image (H1, β = 0.317, CR = 3.134, $p < 0.01$) statistically. Thus, hypothesis 1 has been supported. We tested the hypothesis 2, the ethical responsibility of the company has not found to be meaningful on the corporate image (H2, β = 0.176, CR = 1.737, $p > 0.05$) in terms of statistics. For that reason, hypothesis 2 has not been supported. We tested the hypothesis 3, the legal responsibility fulfillment has found to be meaningful on corporate image (H3, β = 0.216, CR = 1.974, $p < 0.05$) statistically. For that reason, hypothesis 3 has been supported. When we tested hypothesis 4, the discretionary responsibility fulfillment was found to have a meaningful impact statistically in terms of the corporate image (H4, β = 0.218, CR = 2.362, $p < 0.05$). For that reason, hypothesis 4 was supported. After we tested the hypothesis 5, it was found that the impact of corporate image on the behavioral intention (H5, β = 0.867, CR = 12.936, $p < 0.001$) was meaningful statistically. For that reason, hypothesis 5

was supported. The impact of the social responsibility on corporate image was large in the order of economic (β = 0.317), charitable (β = 0.218), and legal (β = 0.216) responsibilities.

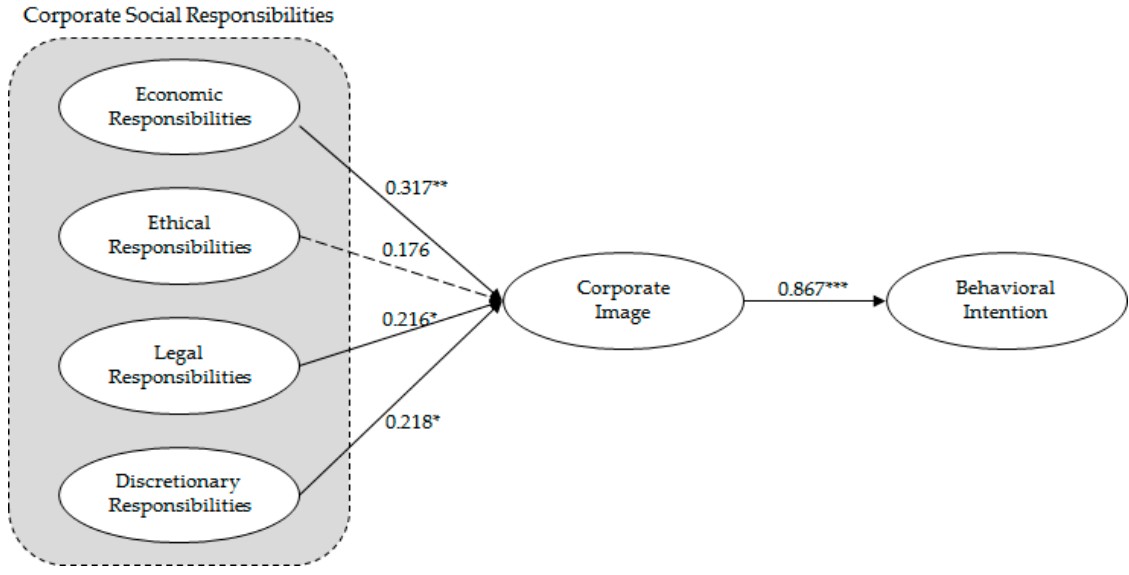

**Figure 2.** The results of the research model (* $p < 0.05$, ** $p < 0.01$, *** $p < 0.001$.).

## 5. Conclusions

The purpose of this investigation is to understand the impact of corporate social responsibility activities on corporate image, and the influence of the corporate image on consumer behavioral intentions by targeting franchise coffee companies. After analyzing the results, among the corporate social responsibilities, the economic, legal, and charitable responsibilities were discovered to have a meaningful impact on corporate image. For that reason, hypotheses 1, 3, and 4 were supported. In addition, the influence of corporate image on consumer behavioral intentions was found to be statistically meaningful. For that reason, the fifth hypothesis was supported.

### 5.1. Theoretical and Managerial Implications

Based on the key findings, a couple of theoretical points could be made.

Firstly, CSR is a critical factor in establishing a positive corporate image. In addition, the companies should once again recognize the importance of their CSR to acquire credibility for their company's image. Although ethical responsibility was found not to be meaningful statistically, the companies should highlight the importance of economic responsibility in the first place. In addition, as proposed by Carroll [38], companies should first deal with their economic responsibility, then take on with their legal responsibility, ethical responsibility, and discretionary responsibilities. Considering that a number of companies are focusing on their discretionary responsibilities, this study discusses the fact that companies should work on their economic responsibility, as well as their legal and ethical responsibilities. Regarding such economic responsibilities, companies should establish a system where they react to customer complaints. In addition, companies should improve their products to provide better quality and service to customers.

Second, it has been proven that a corporate image built on CSR affects consumer behavioral intentions. The direct relationship between the corporate image through the CSR and the consumer behavioral intentions has been proven. In fact, social responsibility activities, with the corporate image as a vehicle, lead positive feedback from customers and increase product preferences [62,74]. In addition, it is clear that social responsibility activities, including economic, legal, and discretionary responsibility activities but not including ethical responsibility activities, could promote a positive corporate image. Such results could provide a guideline to franchise coffee brands as to how important

certain social responsibility activities are for managing their corporate image and engaging with consumers. In particular, it is critical that companies need to establish detailed activation plans, considering that such activities would promote social sustainability and would be a genuine investment in a better life for the next generation.

Thirdly, it has been proven that the companies that implement social responsibility activities will have their customers' behavioral intentions enhanced since their corporate image has been improved. Thus, it is likely that when the CSR is in place to promote a corporate image, the customers will display a behavioral response that includes revisiting intentions, positive word-of-mouth (WOM) spreading, and recommendations.

The strategic implications of this investigation along with the theoretical insights are as follows.

Firstly, the demand for CSR is being raised at diverse levels in broader society. In particular, consumer demand is ever more rising, although there is also an increasing number of consumers who try to deliver their own social responsibilities by reducing materialism and consumerism as a result of individual-centered attitudes, consumer boycotts, and shifting their purchase movements and investment activities towards promoting social responsibility outcomes. CSR activities would only receive a positive consumer impression when they are conducted with a sense of integrity; when customers perceive CSR activities as a means for profit, they show a negative response [91]. In addition, "Warm Hearted Capitalism 4.0" sheds a spotlight on the social responsibility that could encourage and lead people, while also respecting the function of the market and encouraging successful people to become more successful, is in line with the integrity of CSR activities in the modern era.

Secondly, the sincerity of CSR is a measure of all of the franchise coffee brands' social responsibility activities, and it sets itself up as a meaningful guideline, positioning itself as the core agenda in the marketing strategy establishment and the sustainable management pursued by ISO 26000.

Thirdly, as a part of the survival strategy in the new era of changes, companies need to develop themselves as corporate citizens that promote a corporate image by engaging in socially responsible activities and generating long-term profits. In addition, companies will have no other option but to carry out their social responsibilities on a corporate strategy level as the concentration of wealth intensifies and as countries become more multi-nationalized. From now on, the companies will be required to take a strategic approach to obtain coexistence profits for their social responsibility activities regardless of the economic benefits for both the companies themselves and for society. The companies should be aware that the CSR is not an essential option, but it is rather a business practice that they need to take on voluntarily with sincerity for their sustainable growth along with social development.

## 5.2. Limitations and Future Research Directions

The future direction of investigation and the limits as to the investigation conducted for this study are as follows. Firstly, this research has measured the CSR features with the four factors proposed in the preliminary research. However, it is believed that more factors will be found to impact on corporate image in later studies. In addition, the factors that constitute a corporate image are determined based on many variables aside from corporate social responsibility, and their characteristics will differ in terms of companies and industries. Research on the factors that are related to such properties in the companies is necessary. Secondly, there are some limits for the generalization of the research findings on franchise coffee brands' social responsibility activities, as it targeted customers at particular franchise coffee shops. In order to conduct a more objective CSR study, a broader range of measurement targeting a number of general companies is needed. In addition, the direction of social responsibility activities will differ among industries, and it is believed that more stretched-out research should be conducted regarding this. Likewise, in the later research, comparative research on the socially responsible activities of domestic and multinational franchises of coffee companies should be conducted to provide insight into promoting domestic companies' social responsible activities and to establish strategic approach for globalization of these companies.

**Author Contributions:** J.-B.C. has been responsible for data analysis and drafted the manuscript. M.-N.J. supervised the overall research and critically revised the manuscript. All authors read and approved the final manuscript for submission.

**Funding:** The paper was supported by the research grant of the University of Suwon in 2016.

**Conflicts of Interest:** The authors declare no conflict of interest.

## Abbreviations

The following abbreviations are used in this manuscript:

CSR Corporate Social Responsibility

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
