# Peer review of "The Effect of the Corporate Social Responsibility of Franchise Coffee Shops on Corporate Image and Behavioral Intention"

_sustainability, doi:10.3390/su11236849_

Round 1
Reviewer 1 Report
Dear Authors, congratulations with the paper. It very interesting, though the topic is not new. However the findings still give us another point of view when it comes to the CSR topic.
As I wrote, the paper focuses on an important and relevant topic to the journal. While the calibre of the manuscript is evident, there are several issues remain to be carefully addressed before it is considered for publication.
Firstly, You need to improve the abstract with the sentences about the main goal that You wanted achieve and the main hypothesis.
Second issue is that the discussion section despite in general being strong, does not clearly address the ‘so-what’ question—meaning you have done these analysis but what does it actually mean for those Coffee Shops?
Good luck at the revisions and I look forward to read the revised version of the work.
Author Response
The authors greatly appreciate the insightful recommendations and comments of the reviewers. They have made some valuable suggestions that have led to big improvements to the manuscript. All comments of the reviewers have been carefully considered and accounted for in the new version of the manuscript. Moreover, the authors conducted spell check, updated all citations, and proof-read the entire manuscript. Point-by-point the details of the revisions in the manuscript were submitted in separate files.
We hope you concur with us regarding the timeliness and value of this manuscript.
----------------------------------------------------------------------------------------
Dear Authors, congratulations with the paper. It very interesting, though the topic is not new. However the findings still give us another point of view when it comes to the CSR topic.
As I wrote, the paper focuses on an important and relevant topic to the journal. While the calibre of the manuscript is evident, there are several issues remain to be carefully addressed before it is considered for publication.
Point 1: Firstly, You need to improve the abstract with the sentences about the main goal that You wanted achieve and the main hypothesis.
Response 1: Please read the abstract again. I rewrote the abstract to provide my response for Point 1. (in red)
Point 2: The Second issue is that the discussion section despite in general being strong, does not clearly address the ‘so-what’ question—meaning you have done these analysis but what does it actually mean for those Coffee Shops?
Response 2: Please read the 5. Conclusions part. I rewrote discussion to provide my response for Point 2. (in red)
Good luck at the revisions and I look forward to read the revised version of the work.
The authors greatly appreciate the insightful recommendations and comments of the reviewers. You have made some valuable suggestions that have led to big improvements to the manuscript. All comments of the reviewers have been carefully considered and accounted for in the new version of the manuscript. Moreover, the authors conducted spell check, updated all citations, and proof-read the entire manuscript.
I really appreciate your detailed comment. Thank you.

Reviewer 2 Report
Referee Report
Main Comments and Suggestions
You should clarify the contributions of the paper which are not elaborated well in the current paper. You can talk about the following contributions: What insights can you provide based on your finding? Do they push forward our understanding? What should we do with your research? Do you have any suggestions to improve the current regulation or practice? Adding the above discussion and extend your literature review may help you make more contributions and position your contributions better.
The paper seems to claim causality but does not discuss the potential endogeneity issue. For example, the reverse causality such as in Li, F., Young, B., Morris, T. 2019. Corporate Visibility in Print Media and Corporate Social Responsibility. Sustainability forthcoming. You need to discuss this aspect of the story.
Related to the above point, you should study and rationalize the use of firm size measures in the literature since frim size is the key variable in this area and they affect how firms consider their CSR and corporate image. See Dang et al. 2018. Measuring Firm Size in Empirical Corporate Finance. Journal of Banking & Finance, 86:159-176. After all it is the most significant variable in most studies alike. You need to discuss and justify your firm size measure. The results may not be robust to different measures of firm size, which is very common in this area.
Additionally, you should refer to more recent development in this area. See the new literature on how CSR affect financial performance like Li, Minor, Wang and Yu. 2019. A Learning Curve of the Market: Chasing Alpha of Socially Responsible Firms. Journal of Economic Dynamics and Control forthcoming. You need to discuss the motivation to be socially responsible, to give readers a more comprehensive view and a richer story and/or point out future research direction from these perspectives.
You need to seriously proofread the paper and extend and update your references.
In conclusion, I would like to thank the authors for a very interesting, unique and potentially important paper. Hope these comments and suggestions can help further their study.
Author Response
The authors greatly appreciate the insightful recommendations and comments of the reviewers. They have made some valuable suggestions that have led to big improvements to the manuscript. All comments of the reviewers have been carefully considered and accounted for in the new version of the manuscript. Moreover, the authors conducted spell check, updated all citations, and proof-read the entire manuscript. Point-by-point the details of the revisions in the manuscript were submitted in separate files.
We hope you concur with us regarding the timeliness and value of this manuscript.
-----------------------------------------------------------------------------------------
Response to Reviewer 2 Comments
Point 1:
You should clarify the contributions of the paper which are not elaborated well in the current paper. You can talk about the following contributions: What insights can you provide based on your finding? Do they push forward our understanding? What should we do with your research? Do you have any suggestions to improve the current regulation or practice? Adding the above discussion and extend your literature review may help you make more contributions and position your contributions better.
Response 1: Please read the 5. Conclusions part. I rewrote discussion to provide my response for Point 1. (in red)
Point 2:
The paper seems to claim causality but does not discuss the potential endogeneity issue. For example, the reverse causality such as in Li, F., Young, B., Morris, T. 2019. Corporate Visibility in Print Media and Corporate Social Responsibility. Sustainability forthcoming. You need to discuss this aspect of the story.
Response 2: Please read 2. Theory and Hypotheses 2.1 Corporate Social Responsibility. I added literature review related to visibility written by Li, Morris and Young(2019) , 6. Limitations and Future Research Directions and added References as number [43] to provide my response for Point 2. (in blue)
Point 3:
Related to the above point, you should study and rationalize the use of firm size measures in the literature since frim size is the key variable in this area and they affect how firms consider their CSR and corporate image. See Dang et al. 2018. Measuring Firm Size in Empirical Corporate Finance. Journal of Banking & Finance, 86:159-176. After all it is the most significant variable in most studies alike. You need to discuss and justify your firm size measure. The results may not be robust to different measures of firm size, which is very common in this area.
Response 3: Please read 2. Theory and Hypotheses 2.1 Corporate Social Responsibility. I added literature review related to firm size written by Dang, Li and Yang(2018) , 6. Limitations and Future Research Directions and added References as number [44] to provide my response for Point 3. (in blue)
Point 4:
Additionally, you should refer to more recent development in this area. See the new literature on how CSR affect financial performance like Li, Minor, Wang and Yu. 2019. A Learning Curve of the Market: Chasing Alpha of Socially Responsible Firms. Journal of Economic Dynamics and Control forthcoming. You need to discuss the motivation to be socially responsible, to give readers a more comprehensive view and a richer story and/or point out future research direction from these perspectives.
Response 4: Please read 2. Theory and Hypotheses 2.1 Corporate Social Responsibility. I added literature review on how CSR affect financial performance written by Li, Minor, Wang and Yu(2019) , 6. Limitations and Future Research Directions and added References as number [45] to provide my response for Point 4. (in blue)
Point 5:
You need to seriously proofread the paper and extend and update your references.
Response 5: I asked to do proof-read the entire manuscript by native speaker who works for the University of Suwon and she is a professor for the English subject to provide my response for Point 5.
In conclusion, I would like to thank the authors for a very interesting, unique and potentially important paper. Hope these comments and suggestions can help further their study.
Good luck at the revisions and I look forward to read the revised version of the work.
The authors greatly appreciate the insightful recommendations and comments of the reviewers. You have made some valuable suggestions that have led to big improvements to the manuscript. All comments of the reviewers have been carefully considered and accounted for in the new version of the manuscript. Moreover, the authors conducted spell check, updated all citations, and proof-read the entire manuscript.
I really appreciate your detailed comment. Thank you.

Round 2
Reviewer 2 Report
congratulations on the well executed revision.